Negation and uncertainty detection in clinical texts written in Spanish: a deep learning-based approach

Solarte Pabón Oswaldo oswaldo.solartep@alumnos.upm.es 1 2
Montenegro Orlando 2
Torrente Maria 3
Rodríguez González Alejandro 1
Provencio Mariano 3
Menasalvas Ernestina 1
1 Centro de Tecnología Biomédica, Universidad Politécnica de Madrid , Madrid , Spain
2 Escuela de Ingeniería de Sistemas y Computación, Universidad del Valle , Cali , Colombia
3 Hospital Universitario Puerta de Hierro , Madrid , Spain
Ventura Sebastian
Electronic publication date: 2022 Mar 7
Publication date: 2022
Volume: 8
Electronic Location ID: e913
Received 2021 Oct 26; Accepted 2022 Feb 10
Copyright: ©2022 Solarte Pabón et al.
Copyright year: 2022
Copyright holder: Solarte Pabón et al.
License: This is an open access article distributed under the terms of the Creative Commons Attribution License, which permits unrestricted use, distribution, reproduction and adaptation in any medium and for any purpose provided that it is properly attributed. For attribution, the original author(s), title, publication source (PeerJ Computer Science) and either DOI or URL of the article must be cited.
License URL: https://creativecommons.org/licenses/by/4.0/

Keywords: Negation and Uncertainty detection, Deep learning, Text mining, Natural Language Processing, Clinical texts

Funding: European Union’s Horizon 2020 research and innovation program 875160 CLARIFY (Cancer Long Survivors Artificial Intelligence Follow Up) This paper is supported by European Union’s Horizon 2020 research and innovation program under grant agreement No. 875160, project CLARIFY (Cancer Long Survivors Artificial Intelligence Follow Up). The funders had no role in study design, data collection and analysis, decision to publish, or preparation of the manuscript.

==============================
Detecting negation and uncertainty is crucial for medical text mining applications; otherwise, extracted information can be incorrectly identified as real or factual events. Although several approaches have been proposed to detect negation and uncertainty in clinical texts, most efforts have focused on the English language. Most proposals developed for Spanish have focused mainly on negation detection and do not deal with uncertainty. In this paper, we propose a deep learning-based approach for both negation and uncertainty detection in clinical texts written in Spanish. The proposed approach explores two deep learning methods to achieve this goal: (i) Bidirectional Long-Short Term Memory with a Conditional Random Field layer (BiLSTM-CRF) and (ii) Bidirectional Encoder Representation for Transformers (BERT). The approach was evaluated using NUBES and IULA, two public corpora for the Spanish language. The results obtained showed an F-score of 92% and 80% in the scope recognition task for negation and uncertainty, respectively. We also present the results of a validation process conducted using a real-life annotated dataset from clinical notes belonging to cancer patients. The proposed approach shows the feasibility of deep learning-based methods to detect negation and uncertainty in Spanish clinical texts. Experiments also highlighted that this approach improves performance in the scope recognition task compared to other proposals in the biomedical domain.

Introduction

Narrative medical records can provide valuable information to support clinical research, but frequently this information contains uncertain and negated findings (Vincze et al., 2008). Detecting negation and uncertainty is important for medical text mining applications because extracted findings can be incorrectly identified as real or factual events. However, due to the complexity of natural language, automatic identification of negated and uncertain events in clinical texts is not an easy task (Agarwal & Yu, 2010a; Agarwal & Yu, 2010b). Moreover, clinical texts are written by highly skilled physicians and nurses using domain-specific terms, under time pressure, with a rich and complex jargon, which makes these texts differ from those of other domains (Dalianis, 2018a).

Negation changes the meaning of an affirmative sentence, phrase, or word in a negative way. While uncertainty is used to describe ambiguous or suspected events where their truth value cannot be determined due to a lack of information (Jean et al., 2016; Szarvas et al., 2012). In the medical field, it must be known whether the patient definitely suffers, probably suffers, or does not suffer from an illness (Vincze, 2014). In the sentence “A 74-year-old patient with suspected lung carcinoma.”, the truth value of the clinical finding “lung carcinoma” cannot be confirmed, as this finding is uncertain, suspicious, or speculative. Uncertainty detection has also been studied in terms of modality, and it involves related concepts such as subjectivity, hedging, and speculation (Morante & Sporleder, 2012; Cruz Díaz et al., 2012; Solarte Pabón et al., 2021). Furthermore, uncertainty is inherent in many medical decisions, as physicians face uncertain results when they are diagnosing or treating patients (Nikfarjam, Emadzadeh & Gonzalez, 2014). The breadth and complexity of possible diagnoses in medical practice make uncertainty very common in clinical narratives (Alam et al., 2017; Bhise et al., 2018). Consequently, both negation and uncertainty detection are crucial tasks for information extraction in the medical domain.

Negation and uncertainty detection is commonly divided into two sub-tasks: (i) cue identification and (ii) scope recognition. Cues are words or terms that express negation (e.g., not, without, denies) or uncertainty (e.g., possible, probable, suggest) (Cruz Díaz & Maña López, 2019). The scope is the text fragment affected by the corresponding cue in a sentence (De Albornoz et al., 2012). In the sentence: “Probablelung carcinoma with high fever since yesterday, biopsy test will be taken on 25-07-2018.”, the cue is shown in bold and the scope is underlined.

The natural language processing (NLP) community has paid considerable attention to uncertainty and negation detection (Farkas et al., 2010, Morante & Blanco, 2012). Moreover, several corpora annotated for negation and uncertainty have been proposed in the biomedical domain (Vincze et al., 2008; Vincze, 2010b; Uzuner, Zhang & Sibanda, 2009). However, most of these proposals have focused on the English language, while information extraction in the medical domain represents its own challenges in languages other than English (Névéol et al., 2018a). Most proposals developed for medical texts written in Spanish (Santiso et al., 2018; Santiso et al., 2020; Cotik et al., 2016; Costumero et al., 2014), have been focused only on negation detection. Uncertainty detection for Spanish medical texts has not yet been sufficiently addressed and can be improved.

Negation and uncertainty detection have been widely addressed using rule-based approaches (Chapman et al., 2001; Harkema et al., 2009; Kesterson et al., 2015), and classical machine learning-based approaches (Cruz Díaz et al., 2012; Morante & Daelemans, 2009b; Jimnez-Zafra et al., 2021). Rule-based methods can suffer from a lack of flexibility and universality (Zhou et al., 2018). While classical machine learning methods depend on hand-crafted features, and they often require a complex and time-consuming feature engineering process and analysis to obtain a good performance (Minaee et al., 2020). Recently, deep learning approaches have been shown to improve performance at processing natural language texts in several tasks such as named entity recognition (NER) (Lample et al., 2016), question answering (Bordes, Chopra & Weston, 2014), and language translation (Sutskever, Vinyals & Le, 2014). One of the advantages of deep learning approaches is they can automatically learn features from data, instead of adopting handcrafted features. Embedding models such as Word2Vec (Mikolov et al., 2013), Glove (Pennington, Socher & Manning, 2014), and FasText (Bojanowski et al., 2017) have been popularly used in text processing applications. These models also have been applied in the biomedical field (Wang et al., 2018; Soares et al., 2019). Moreover, the development of contextual embeddings (Peters et al., 2018) and transformer-based models (Devlin et al., 2019) have shown that such representations are able to improve performance on a wide range of natural language processing tasks (Liu, Kusner & Blunsom, 2020; Gu et al., 2020; Pires, Schlinger & Garrette, 2019). Recently, deep learning-based approaches have also been proposed for negation and uncertainty detection (Dalloux, Claveau & Grabar, 2019; Britto & Khandelwal, 2020; Fei, Ren & Ji, 2020; Zavala & Martinez, 2020), showing promising results that will allow to advance in the field.

Motivated by improvements in deep learning methods to process natural language texts, in this paper we propose an approach for negation and uncertainty detection in clinical texts written in Spanish. This approach explores two deep learning methods to perform negation and uncertainty detection: (i) Bidirectional Long-Short Term Memory with a Conditional Random Field layer (BiLSTM-CRF) and (ii) Bidirectional Encoder Representation for Transformers (BERT). The proposed approach takes advantage of transfer learning techniques to perform uncertainty and negation detection in clinical texts. Transfer learning aims to transfer knowledge from pre-trained resources to improve the performance on a new target task (Liu et al., 2019a; Peng et al., 2020; Ortiz Suarez, Romary & Benoit, 2012; Panigrahi, Nanda & Swarnkar, 2021). In this approach we exploit pre-trained resources such as word embeddings (Soares et al., 2019; Mikolov et al., 2013) and contextualized embeddings (Devlin et al., 2019) to perform negation and uncertainty detection as a sequence labeling task. The most significant contributions of this paper are:

• A deep learning-based approach for negation and uncertainty detection in clinical texts written in Spanish. The main advantage of this approach is the use of word embeddings and contextual embeddings to automatically represent text features, avoiding the time-consuming feature engineering process. In the Spanish language, most of the previous studies have focused only on negation detection. Meanwhile, our approach goes further and in addition to negation, it also performs uncertainty detection. Code developed in this approach is public accessible from GitHub (https://github.com/solarte7/NegationAndUncertainty).

• Exploiting transfer learning techniques to perform negation and uncertainty detection in Spanish clinical texts. In particular, transfer learning exploitation is applied in two ways: (i) Creating pre-trained clinical embeddings and (ii) Fine-tuning the BERT model. The generated clinical embeddings improve the performance of the BiLSTM-CRF neural model for detecting uncertainty and negation. Furthermore, the BERT model is fine-tuned with a classification layer on top. To the best of our knowledge, this is the first approach that uses a transformer-based method to perform both uncertainty and negation detection in clinical text written in Spanish.

• Improvement of performance in the scope recognition task, compared to other proposals in the biomedical domain for the Spanish language. Results obtained with both BiLSTM-CRF and BERT models have shown an improvement over other studies. Performed tests were evaluated using NUBES (Lima Lopez et al., 2020) and IULA (Marimon, Vivaldi & Bel, 2017), two public corpora for the Spanish language. Obtained results in the scope recognition task have shown an F-score of 92% and 80% for negation and uncertainty detection, respectively.

• Validation of the proposed approach with a real-life dataset which contains annotations of patients diagnosed either with lung or breast cancer. To perform this validation, we used trained models on the NUBES corpus (Lima Lopez et al., 2020) to predict negation and uncertainty in this new dataset. The validation process shows the ability of deep learning-based models to predict negation and uncertainty in a different dataset to the one they were trained on.

The remainder of this paper is organized as follows: ‘Related works’ shows previous studies about uncertainty and negation detection in the biomedical domain. In ‘Materials and methods’ datasets and proposed methods to perform negation and uncertainty detection are described. ‘Experimentation and Results’ explains the experiments carried out to validate our approach, and ‘Discussion’, provides a discussion of the results obtained. Finally, ‘Conclusions’ includes conclusions and future work.

Related works

The high percentage of uncertain and negated sentences within clinical texts has motivated more research on this field. In particular, 12% of the sentences contained in Medline abstracts are uncertain, and 20% are negated (Vincze, 2014). Several annotated corpora have been proposed in the biomedical domain (Vincze et al., 2008; Vincze, 2010b; Uzuner, Zhang & Sibanda, 2009). In the Bioscope corpus (Vincze et al., 2008), 18% of the sentences contain uncertain findings (Szarvas et al., 2012). Moreover, Jiménez-Zafra et al. (2020b) describes a comprehensive review of different corpora annotated with negation. This review includes the analysis of several corpus in different languages including English and Spanish.

In what follows, we will address the three approaches that have been found in the literature for negation and uncertainty detection: (i) rule-based, (ii) machine learning-based, and (iii) deep learning-based.

Rule-based approaches

Rule-based approaches use declarative methods for creating manually crafted rules that extract uncertain and negated findings. One of the most widely used rule-based algorithm to detect negation in medical records is NegEx (Chapman et al., 2001). This algorithm has been recognized as one of the most useful approaches for the detection of negated medical concepts. However, several studies have been proposed to improve NegEx’s performance (Elazhary, 2017; Harkema et al., 2009), Deepend2015. In particular, the ConText algorithm (Harkema et al., 2009) extends NegEx for determining whether clinical conditions mentioned in clinical reports are negated, hypothetical, historical, or experienced by someone other than the patient. The proposal described in Wu et al. (2011) also extends the NegEx algorithm to detect uncertainty by adding a separate category of uncertainty terms. In  Velupillai et al. (2014), the authors proposed ConTextSwe, an adaptation of the ConText algorithm to Swedish. Several proposals have also been presented to adapt the NegEx algorithm to Spanish (Costumero et al., 2014; Stricke, Ignacio & Cotik, 2015; Santamaria, 2019), but these proposals have only focused on negation detection.

The above-mentioned proposals use a similar approach to recognize the scope; that is the search for a termination term in a lexicon which indicates the end of the scope. However, one disadvantage of this approach occurs when the sentence does not contain any termination term. In these cases, the scope recognition fails because all tokens in a sentence can be taken as the scope. To deal with this problem, other studies have included the use of syntactic properties of the sentence to extract the scope (Cotik et al., 2016; Zhou et al., 2015; Peng et al., 2018). Although rule-based approaches have been widely used in the biomedical domain, their main disadvantages are the large amount of time it takes to create rules manually, and the lack of flexibility and universality (Zhou et al., 2018).

Machine learning-based approaches

In machine learning-based approaches, negation and uncertainty detection is formulated as a classification problem where both cues and scope detection are considered as sequence labeling tasks. These approaches commonly follow two steps: in the first step, hand-crafted features are extracted from documents and, in the second step those features are used to train a classifier to perform predictions. Early machine learning-based proposals deal only with recognizing negation and uncertainty at the sentence level (Clausen, 2010; Skeppstedt, Paradis & Kerren, 2016; Velupillai, Dalianis & Kvist, 2011; Shaodian et al., 2016). In these cases, the complete sentence is considered an uncertain or negated fact. However, it is necessary to identify what tokens in the sentence are affected and which are not. One of the firsts machine learning proposals which deal with scope recognition is described in  Morante & Daelemans (2009b). This study uses four classifiers for negation detection using approaches such as Support Vector Machines (SVM) and Conditional Random Fields (CRF) (Lafferty, McCallum & Pereira, 2001). The approach was evaluated using the Bioscope corpus (Vincze et al., 2008) and obtained an F1-score of 80.4% in the scope recognition task. In  Morante & Daelemans (2009a), the above proposal was extended to deal with uncertainty detection.

Another proposal for negation detection is described in  Agarwal & Yu (2010a). In this work, negation cues and their scope are detected in clinical notes and biological literature using CRF as a machine learning algorithm. With the goal of detecting both negation and uncertainty,  Cruz Díaz et al. (2012) proposed a two phases machine learning model. The first phase classifies the cues, and the second predicts the scope. Reported results showed an F1-score of 91% and 72% for negation and uncertainty scope, respectively. All the machine learning-based proposals mentioned above use the BioScope corpus (Vincze et al., 2008), which is focused on the English language. In the case of the Spanish language,  Santiso et al. (2018) proposed a CRF-based classification model for negation detection in clinical records. This study was evaluated using IULA (Marimon, Vivaldi & Bel, 2017), a corpus annotated with negation in clinical text written in Spanish, and obtains an F1-score of 81% in scope recognition.

Although classical machine learning-based approaches have addressed limitations of rule-based methods, one disadvantage of these approaches is the reliance on the hand-crafted features that require tedious, time-consuming feature engineering along with analysis to obtain good performance (Minaee et al., 2020). This fact can be seen in studies such as the ones reported in Jiménez-Zafra et al. (2020a) and Jiménez-Zafra et al. (2020b), where a CRF-based system is trained with a considerable set of hand-crafted features to perform negation detection.

Deep learning-based approaches

The core component of these approaches is the use of word embedding models that map a set of texts into a low-dimensional continuous space (Mikolov et al., 2013; Pennington, Socher & Manning, 2014; Bojanowski et al., 2017). Contextualized embeddings such as ELMO (Peters et al., 2018) and BERT (Devlin et al., 2019) have also shown that such representations are able to improve performance on sequence labeling tasks. Recurrent neural networks(RNN) (Goldberg & Hirst, 2017; Hochreiter & Schmidhuber, 1997), and Convolution Neural Networks(CNN) (Lopez & Kalita, 2017) have also been used to process text in the biomedical domain.

In  Qian et al. (2016), the authors proposed a Convolutional Neural Network-based model to extract negation and uncertainty scope from biomedical texts written in English. This model first extracts path and position features from syntactic trees with a convolutional layer whose features are concatenated into one feature vector. This vector is finally fed into the softmax layer to obtain the output vector. Tests and validation are conducted using the Bioscope corpus, showing the ability of the deep neural approaches to deal with negation and uncertainty detection. In  Fancellu et al. (2017) is proposed a Bidirectional Long-Short Term Memory (BiLSTM) model to extract negation scope from English and Chinese texts. The performance shows an F-score of 89% for English and 79% for Chinese using the CNeSp corpus (Zou, Zhou & Zhu, 2016). The conclusion of that research is a suggestion to use more training data to make progress on negation detection. In  Taylor & Harabagiu (2018), the authors proposed a BiLSTM neural model for negation detection from electroencephalography reports written in English. Reported results show an F-score of 88% for the scope recognition task using the Bioscope corpus. In  Bhatia, Celikkaya & Khalilia (2018), an encoder–decoder neural architecture that combines a shared encoder and different decoding schemes to jointly extract entities and negations is proposed. Reported results show 90% in F-score for negation detection using data from the 2010 i2b2/VA challenge task (Uzuner et al., 2011).

An attention mechanism to perform uncertainty cue identification using data from CoNLL2010 shared task (Farkas et al., 2010) is developed in  Adel & Schütze (2017). Reported results have shown that combining attention layers with RNN and CCN models increases the performance for uncertainty detection in the English language, showing 85% in F1-score. The attention layer helps the model to recognize which part of the input data is important during the training, allowing the networks to focus on specific information by generating a weight vector. Another proposal that combines an attention layer with an RNN for detecting negated, possible and hypothetical medical findings from clinical notes is described in Chen (2019). In  Khandelwal & Sawant (2020), the authors proposed NegBert, a model for negation detection using BERT contextual embeddings. NegBert has been trained for the English language using three corpora from different domains, including the BioScope corpus for the biomedical domain (Vincze et al., 2008). Reported results have shown an improvement in the scope resolution task with 93% in F1-score and comparable results in the cue identification task. In  Shaitarova, Furrer & Rinaldi (2020b), the authors extended the  Khandelwal & Sawant (2020) proposal to deal with both uncertainty and negation detection using transformer-based architectures such as BERT (Devlin et al., 2019), XLNet (Yang et al., 2019) and Roberta (Liu et al., 2019b). This study was also focused on the English language using the Bioscope corpus and the SFU Review corpus (Konstantinova et al., 2012).

In recent years, the interest in processing negation and uncertainty has also grown in languages other than English. In  Dalloux, Claveau & Grabar (2019), a BiLSTM-CRF model focused on the French language is presented. The approach was validated using a dataset from the National Cancer Institute in France (NCI (https://en.e-cancer.fr/)) and obtained an F-score of 90% and 86% for negation and uncertainty scope detection, respectively. Al-khawaldeh (2019) proposed an Attention-based BiLTSM model to perform speculation detection in Arabic medical texts. This proposal obtained an F-score of 73.5% in the scope recognition task using the BioArabic corpus (Al-khawaldeh, 2016).

In the case of the Spanish language,  Santiso et al. (2018) proposes a CRF-based model combined with word embeddings to perform negation scope detection. That proposal depends on several hand-crafted features to increase the performance of the proposed model. In  Santiso et al. (2020) is described a BiLSTM-based proposal that uses word embeddings and lexical properties of the sentence such as word-forms and lemmas to identify negated medical concepts. This proposal uses the IxaMed-GS corpus (Oronoz et al., 2015), a private dataset consisting of 75 clinical notes written in Spanish. Reported results showed 83% in F-score for the scope recognition task. Zavala & Martinez (2020) evaluate the impact of embeddings on different deep learning-based models to perform negation and speculation detection using several corpora from English and Spanish. In particular, for the Spanish language, the performed tests were conducted using IULA (Marimon, Vivaldi & Bel, 2017) and SFU ReviewSP-NEG (Jiménez-Zafra et al., 2018), two corpora focused only on negation annotation. Moreover, the BiLSTM-based approach proposed in  Zavala & Martinez (2020) uses general domain word embeddings for Spanish (Cardellino, 2019), and several annotated features such as Part of Speech (POS) tagging and information about overlapping entities to perform negation detection. The above proposals for Spanish share two features: (i) they perform only negation detection but not uncertainty detection. (ii) They use hand-crafted features, syntactic properties, or lexical information of the sentence to detect negation in clinical documents.

The first study that can be found for Spanish to perform negation and uncertainty detection is the one presented in  Lima Lopez et al. (2020) in which the NUBES corpus is presented. This corpus contains both negation and uncertainty annotations from clinical notes written in Spanish. Moreover, in that proposal, the authors provide a BiLSTM-based model whose results showed a 90% and 78% in F1-score for negation and uncertainty scope detection, respectively. Although this study has shown promising results, the main disadvantage is that it relies on several hand-crafted features and syntactic properties of the sentence to feed the system, making the feature engineering process time-consuming.

Table 1 shows a summary of the most relevant deep learning-based approaches to perform negation and uncertainty detection. From this table, it is important to highlight the following facts: (i) Most of the proposals have focused on the English language, as a result of the availability corpora (Vincze et al., 2008; Uzuner et al., 2011; Farkas et al., 2010); (ii) Most of the existing proposals developed for the Spanish language concentrate only on negation detection, (iii) Although the proposal described in Lima Lopez et al. (2020) aims to detect uncertainty and negation in Spanish, its main weakness is the dependence on hand-crafted features. These facts suggest that uncertainty and negation detection in Spanish can be improved.

Table 1 Summary of Deep learning-based approaches.

Proposal	Approach	Language	Corpus	Negation	Uncertainty	
Qian2016	CCN	English	Bioscope	yes	yes	
Fancellu2017	BiLSTM	English & Chinese	Bioscope & CNeSp	yes	no	
Taylor2018	BiLSTM	English	BioScope	yes	no	
Uzuner2011	Encoder- Decoder	English	i2b2/VA NLP challenge	no	yes	
Attention2017	RNN + Attention	English	Bioscope	yes	yes	
Attention_Chen_2019	RNN + Attention	English	i2b2/VA NLP challenge	yes	no	
NegBERT2020	BERT	English	Bioscope	yes	no	
Dalloux2019	BiLSTM	French	NCI - France	yes	yes	
Al-khawaldeh2019	Attention + BiLSTM	Arabic	Bio Arabic	no	yes	
Santiso2018	Embeddings + CRF	Spanish	IULA	yes	no	
Santiso2020	BiLSTM	Spanish	IxaMed-GS	yes	no	
Zavala2020	BiLSTM, BERT	Spanish	IULA	yes	no	
Lima2020	BiLSTM	Spanish	NUBES	yes	yes	

Thus, in this paper, we propose a deep learning-based approach that also relies on BERT and BiLSTM as other proposals reviewed in the literature. However, the proposed approach in this paper presents the following differences and advantages over other proposals for the Spanish language:

• We propose using suitable biomedical word embeddings and contextual embeddings for Spanish, which can represent text features automatically. In our approach, we only use information from word embeddings and contextual embeddings to train deep learning-based models. These embeddings allow us to obtain competitive results for both uncertainty and negation detection, which avoids relying on a manual feature engineering process. In addition, this approach does not depend on the syntactic or lexical properties of the sentence.

• We compare the performance of the BiLSTM and BERT models analyzing the advantages and shortcomings of each model. To the best of our knowledge, this is the first study that compares the performance of these models for uncertainty and negation detection in Spanish.

• We perform a validation process in an external dataset that is not used to train the deep learning-based models. Previous approaches for the Spanish language only report results for the same corpus used in the training process. The proposed approach uses trained models on the NUBES corpus (Lima Lopez et al., 2020) and validates them in the Cancer dataset section ‘Datasets’. It is important to note that this validation has been required to annotate a new corpus (The Cancer dataset) which is a time-consuming task. In summary, our approach leverages the NUBES corpus to train models and use them to detect uncertainty and negation in the Cancer dataset. To our knowledge, this is the first study that reports results obtained after applying models trained with the NUBES corpus in another dataset. The results obtained show the potential of models trained this way to be applied to detect negation and uncertainty on external datasets in the biomedical domain.

Materials and Methods

In this section, we first show the datasets used for training, testing, and validating the proposed approach. Next, we will describe the deep learning-based methods for negation and uncertainty detection.

Datasets

NUBES (Lima Lopez et al., 2020) and IULA (Marimon, Vivaldi & Bel, 2017) are two public corpora available for the Spanish language that will be used to train models. Additionally, an in-house annotated dataset with real-life data of cancer patients was manually annotated and will be used for validation purposes. Details about each dataset are given as follows:

• NUBES (https://github.com/Vicomtech/NUBes-negation-uncertainty-biomedical-corpus): a public corpus which consists of 29,682 sentences obtained from anonymized health records annotated with negation and uncertainty (Lima Lopez et al., 2020). NUBES is the largest publicly available corpus for negation in Spanish clinical records, and the first corpus that also incorporates the annotation of uncertainty. This corpus contains annotations for syntactic, lexical, and morphological negation cues.

– Syntactic negation: are cues represented by function words or adverbs, for instance: not, without, never (“no, sin, nunca”)

– Lexical negation: are words or multi-word expressions which indicate negation depending on the context. They include verbs, adjectives or noun phrases, for example: negative, denies, withhold (“negativo, niega, suspender”).

– Morphological Negation: are words which refer to negation by means of affixes, for instance: afebrile, asymptomatic (“Afebril, Asintomático”).

In the case of uncertainty, the NUBES corpus contains annotations for lexical and syntactic cues. Lexical cues are words that express uncertainty depending on the context. Lexical cues include words such as “probable”, “possible”, and “compatible with” (“Probable, posible, compatible con”). Syntactic cues include only the disjunctions words “Versus”, “Vs”, “Or” (“Versus, vs, o”). These cues were only annotated when they appeared by themselves in a context of uncertainty.

IULA (http://eines.iula.upf.edu/brat/#/NegationOnCR_IULA/): a public corpus which contains 3,194 sentences extracted from anonymized clinical records and manually annotated with negation cues and their scopes (Marimon, Vivaldi & Bel, 2017). This corpus was extracted from clinical notes from one hospital in Barcelona (Spain) and contains annotations only with negation. Syntactic negation, and lexical negation cues have been annotated in this corpus, but not morphological negation.

Cancer dataset: an in-house manually annotated dataset with data from patients either suffering with lung or breast cancer. This dataset was extracted from real-life clinical notes belonging to cancer patients from ”Hospital Universitario Puerta de Hierro, Madrid Spain”. The dataset contains 2,700 sentences annotated with both negation and uncertainty. We have received informed consent from participants of our study before annotating this dataset. The Cancer dataset was annotated with syntactic, lexical and morphological negation. In the case of uncertainty, this dataset contains syntactic and lexical cues with their respective scopes. In the Cancer dataset, negation was more frequently found in sentences that describe symptoms, medical tests results, and treatments. On the other hand, uncertainty was more frequently found in sentences that describe cancer diagnosis. Figure 1 shows a set of sentence examples extracted from the Cancer dataset. These sentences show negation and uncertainty cues and their scopes.

Figure 1 Sentences with negation and uncertainty cues and their scopes.

The Cancer dataset was manually annotated by two clinicians using the BRAT annotation tool (https://brat.nlplab.org/). These annotators were guided by a data scientist who prepared documents and explained the BRAT tool functionalities. The annotation process was performed in four steps, as follows:

1. Training: This step aimed to train annotators to manually annotate cues (negation and uncertainty) and their scopes in clinical notes.

2. Pre-annotation: Annotators started with the annotation of 20% of the sentences in the dataset. Once they finished, they met the data scientist to resolve questions regarding the annotation process.

3. Annotation: In this step, the clinicians annotated 100% of the sentences independently and performed the annotation process separately.

4. Disagreement resolution: The annotators met with a linguist who reviewed the cases where there were disagreements and guided the clinicians to resolve them.

The inter-annotator agreement (IAA) measures the quality of annotations and the reliability of an annotated corpus. Although, both Kappa statistic (Cohen, 1960) and F-measure (Hripcsak & Rothschild, 2005) have been used to calculate de IAA between annotators, it has been argued that using the F-measure is more suitable than Kappa in contexts where annotated entities might have different token spans (Campillos-Llanos et al., 2021; Brandsen et al., 2020; Dalianis, 2018b; Pradhan et al., 2015; Ogren, Savova & Chute, 2008). This is because obtaining the Kappa value requires calculating the probability of agreement by chance. This can be very small or close to zero (Oronoz et al., 2015) in contexts such as annotating the negation and uncertainty, as most tokens in the dataset do not receive any annotation.

Thus, in a similar way to the one proposed for the Bioscope corpus (Vincze et al., 2008) and the IxaMed corpus (Oronoz et al., 2015), we calculate the IAA for the Cancer dataset using the F-measure (Hripcsak & Rothschild, 2005). In particular, to calculate the IAA between annotator pairs, one annotator is considered as the gold standard. Precision is the percentage of correct positive annotations made by the second annotator, and Recall is the percentage of positive cases annotated by the second annotator (Alnazzawi, Thompson & Ananiadou, 2014). The F-measure is the harmonic mean between Precision and Recall. We calculate the IAA after the annotation step is finished. Table 2 shows the IAA using the F-measure, which was obtained at token level and entity level, as follows:

Table 2 Inter-annotator agreement for the Cancer dataset.

	Token level IAA	Entity level IAA	
Negation cue	0.96	0.95	
Negation scope	0.94	0.92	
Uncertainty cue	0.94	0.93	
Uncertainty scope	0.92	0.90	

• Token level agreement: the IAA for each token is calculated separately. When a cue or a scope contains several tokens, a pair of annotators can annotate different spans. In this case, correct annotations are those tokens where both annotators agree. For instance, in the sentence “Sindolor toráxico agudo” (No acute chest pain), the scope annotated by one annotator is shown underlined. If the second annotator annotates only the tokens dolor toráxico”, then these two tokens are considered an agreement and the other token is considered a disagreement.

• Entity level agreement: the IAA is measured at cue or scope level. In this case, the correct annotations are those cues or scopes where both annotators agree in all annotated tokens. In the above example, an annotation is considered an agreement when the two annotators annotate the same tokens in the scope “dolor toráxico agudo”. Otherwise, it is considered a disagreement.

Tables 3 and 4 show a descriptive analysis of the datasets previously mentioned. This analysis aims to check how sentences, cues and scopes behave in Spanish clinical texts. The analysis has shown that negation and uncertainty have specific features as follows:

Table 3 A summary of the datasets used in the proposed approach.

Indicators	NUBES	IULA	Cancer dataset	
Number of sentences	29,682	3,194	2,700	
Sentences with negation	25.5%	34%	27%	
Sentences with uncertainty	7.5%	–	12%	
Maximum number of tokens	210	159	181	
Mean (Number of tokens)	18	14	15	
Median (Number of tokens)	14	10	12	
First quartile	9	6	7	
Third quartile	23	19	18	

Table 4 A summary of cues and scopes in the datasets.

Indicators	NUBES	IULA	Cancer dataset	
Number of distinct negation cues	345	46	52	
Number of distinct uncertainty cues	303	–	70	
Total negation cues	9318	1145	804	
Total uncertainty cues	2529	–	345	
Syntactic negation cues	85%	92%	83%	
Lexical negation cues	6%	8%	10%	
Morphological negation cues	9%	7%	2%	
Syntactic uncertainty cues	2%	–	1%	
Lexical uncertainty cues	98%	–	99%	
Continuous scopes	95%	96%	97%	
Discontinuous scopes	5%	4%	3%	

• Sentence length: is the number of tokens in sentences where uncertainty or negation appears. According to Table 3, the analyzed datasets have similar values in indicators such as median, first quartile, and third quartile. In the case of the NUBES corpus, the value of median is 14, which indicates that 50% of the sentences have 14 tokens or fewer. Similar behavior occurs in the IULA corpus and the Cancer dataset, where the median is 10 and 12, respectively. This fact suggests that in clinical texts written in Spanish negation and uncertainty can frequently occur in short sentences. However, there are also large and more complex sentences where several cues can appear as well as negation as uncertainty. In fact, negation and uncertainty can appear in the same sentence. According to Fig. 1, in the fourth example there is a short sentence, while the fifth example shows a large and more complex sentence.

• Cues: are distributed as follows; in the NUBES corpus 85% of negation annotations contain syntactic negation, 6% lexical, and 9% morphological negation. Similar values can be found in the IULA and the Cancer datasets. In the case of uncertainty, 98% of annotations contain lexical uncertainty and only 2% syntactic uncertainty, for the case of the NUBES corpus. Similar values can be found in the Cancer dataset (See Table 4).

• Scopes: can behave in a continuous or discontinuous way. A continuous scope occurs when the tokens affected by a specific negation or uncertainty cue are continuous in the sentence. On the other hand, a discontinuous scope occurs when the sequence of tokens affected by a cue are separated in different positions of the text sentence. In Fig. 1, the first and the fourth sentences contain continuous scopes. In contrast, the third sentence contains a discontinuous scope. According to Table 4, most cases are continuous scopes. In the NUBES corpus, 95% of annotations correspond to continuous scopes and only 5% to discontinuous scopes. Similar behavior can be seen in the IULA and the Cancer datasets. Additionally, the scope can appear after the cue, before the cue, or on both sides. According to Fig. 1, in the fourth sentence the scope appears to the right (after) of the cue “Sin”. In the second sentence, the scope appears to the left (before) of the cue “negativo”. Meanwhile, in the third example, the scope is in both sides of the cue.

Deep learning-based methods for negation and uncertainty detection

The proposed approach addresses negation and uncertainty detection as a sequence-labeling task, where each token in a sentence is classified as being part of the cue or the scope. The BIO (short for Beginning, Inside, Outside) tagging format is used to labeled each token. This approach recognizes cues and their scopes in a single stage for both negation and uncertainty. To perform negation and uncertainty detection from clinical text written in Spanish, we explore two deep learning-based models: BiLSTM-CRF and BERT.

We chose these models as they have shown to be widely used to perform sequence-labeling tasks. In particular, the BiLSTM-CRF model has shown its ability to use previous and future information in a text sequence to predict the state of a current input (Giorgi & Bader, 2019). This ability can be useful to resolve the scope, because in clinical texts written in Spanish, the scope can appear after and before the cue, as was shown in section ‘Datasets’. Thus, analyzing the context that is before and after a cue in the sentence can help in the scope recognition. On the other hand, the BERT model is based on transformer architectures which also have been improved sequence-labeling tasks in the biomedical domain (Lee et al., 2020). Transformers are able to learn long-range dependencies between words using a self attention mechanism (Vaswani et al., 2017). This mechanism can be useful to learn dependencies between cues and their scope.

In this section, we first show how the corpora mentioned in section ‘Datasets’ is pre-processed, and then how the BiLSTM-CRF and BERT models were trained, as follows:

Corpus pre-processing

The corpus pre-processing aims to transform the annotated corpora into a matrix representation of data which will be used to train the deep learning-based models. Pre-processing a corpus includes four steps (See Fig. 2), as follows:

Figure 2 Pre-processing steps to transform an annotated corpus into matrices.

1. Annotations to BIO format: this step generates two matrices (T, L) using the BIO tagging format. The matrix T contains one row for each sentence in the corpus and the columns are the tokens of the sentence. On the other hand, the matrix L contains one row per sentence in which columns represent the BIO labels for each token of the sentence. Let i be the the number of sentences in the corpus, and j be the maximum length of tokens in the sentences. Since the sentences do not have the same number of tokens, padding tokens are added to the matrices.

2. Dictionary creation: in this step two dictionaries (DT, DL) are created to represent tokens and BIO labels using unique identifiers. The dictionary DT contains the set of unique tokens in the corpus and their numeric identification. The dictionary DT contains the set of BIO labels in the corpus. The size of DT is the total number of different tokens in the corpus and the size of DL is the number of different BIO labels in the corpus.

3. Matrix codification according to dictionaries: this step uses the dictionaries DT and DL to generate two new matrices (T′, L′). Matrix T′ contains a numeric representation for each token using the identifications from DT. Matrix L′ contains a numeric representation for each BIO label using the identifications from DL.

4. One hot vector creation: this step generates matrix H which contains a one hot vector representation for each token and its BIO label for each sentence. The size of the matrix H is i × j, where i represents the number of sentences in the corpus, and j is the maximum length of tokens of all sentences. Each position Hi, j contains one hot vector representation which represents the BIO label for the token in Ti, j position.

BiLSTM-CRF

The first model is a Bidirectional Long-Short Term Memory with a CRF layer(BiLSTM-CRF) neural net. This model is based on neural architectures described in Lample et al. (2016); Huang, Xu & Yu (2015); Collobert et al. (2011) and consist of three layers: Embedding layer, BiLSTM layer,and CRF layer (see Fig. 3). To train the BiLSTM-CRF model, the approach uses pre-processed information from matrices T and H section ‘sub-preprocessing’.

Figure 3 Negation and uncertainty detection using the BiLSTM-CRF model.

• Embedding layer: This layer allows the approach to automatically represent text features using dense vector representations. In these vectors, words with a similar context in the text have a similar representation. The Embedding layer enables the approach to represent each word into a fixed length vector of defined size. We use two different word embeddings in this approach:

– Biomedical embeddings: These embeddings were trained using full-text medical papers written in Spanish (Soares et al., 2019). The papers were taken from Scielo (a scientific electronic library (https://scielo.isciii.es/scielo.php)), and a subset of Wikipedia articles related to Pharmacology, Medicine, and Biology.

– Clinical embeddings: We create in-house embeddings trained with more than 1 million clinical notes written in Spanish. These clinical documents were provided in a raw format by two public hospitals in Madrid (Spain) and Cali(Colombia). We use the FasText (Bojanowski et al., 2017) method for creating word embeddings using a vector size of 300 positions by default.

• BiLSTM layer: the Bidirectional LSTM (BiLSTM) layer captures both left and right contexts of words to produce a vector representation of text sequences. Given a sentence (w1, w2, w3, …wn) as input, where n is the number of words, this layer processes the sentence using two steps: As an output, the BiLSTM layer generates a vector representation hi for each word by concatenating the values fi and bi. The output vector hi contains a sequence of probabilities for each label to be predicted. The BiLSTM layer computes the Forward and Backward steps separately. Therefore, the values fi and bi are calculated independently.

– A Forward step processes the sentence from left to right, where each element fi represents the value of the left context for the word wi in the sentence.

– A Backward step processes from right to left where each element bi represents the value of the right context for the word wi in the sentence.

• CRF layer: this layer predicts the label sequence with the highest prediction score from all sequences generated by the BiLSTM layer. Although the BiLSTM layer generates probabilities for each label to be predicted, these probabilities are calculated independently. For sequence labeling tasks it is crucial to consider correlations and dependencies across output labels. Therefore, this layer uses an implementation of the CRF algorithm (Lafferty, McCallum & Pereira, 2001) to improve the predictions for each label. The CRF algorithm considers correlations between other labels and jointly decodes the best chain of labels for a given input text sentence.

Bidirectional Encoder Representation for Transformers (BERT)

BERT (Devlin et al., 2019) uses a transformer-based architecture to learn representation of texts in a bidirectional way by considering both the left and the right context of words. In the proposed approach, the BERT model is fine-tuned with a classification layer on top. We use multilingual BERT as contextualized embeddings to perform negation and uncertainty detection in clinical notes written in Spanish. Multilingual BERT has been pre-trained on data in 104 languages with the same training objectives as BERT: masked language modeling and next-sentence prediction.

Figure 4 shows the process for detecting negation and uncertainty using multilingual BERT. This process uses information from the matrices T and L section ‘Corpus pre-processing’, and processes each sentence using three steps: Tokenization, BERT Processing, and Classification & Post-processing.

Figure 4 Negation and uncertainty detection using multilingual BERT.

1. Tokenization: the goal of this step is to take a raw text sentence as input and tokenize it using a WordPiece Tokenization method (Wu et al., 2016). This method relies on the idea that the most frequent used words should not be divided into smaller sub-words, but rare words should be split into meaningful sub-words. The tokenization method used by BERT was previously trained on a raw text dataset to learn vocabulary with most frequent words. In the example of Fig. 4, the words “muestra” and “células” have not been split into sub-words because the WordPiece tokenization method considers these words as frequent words in its vocabulary. In contrast, the words “biopsia” and “cancerígenas” are considered rare words by the algorithm because they are not in its vocabulary. Consequently, these words have been split in sub-words and the characters “##” are used to separate these tokens. The WordPiece Tokenization algorithm uses a Byte Pair Encoding strategy (Schuster & Nakajima, 2012) to generate new sub-words. This tokenization method aims to improve the handling of rare and unseen words in a dataset by providing a balance between the flexibility of character-delimited tokenizers and the efficiency of word-delimited tokenizers. Finally, in this step, two special tokens are added to the sentence: [CLS] and [SEP]. The [CLS] token always appears at the beginning of a sentence, and the [SEP] token is used to separate segments of a sentence.

2. BERT Processing: In this step, the approach takes the tokenized sentence as input from the previous step and process it as follows:

• An embedding representation (Ei) for each token in the sentence is obtained. This representation is composed of three embeddings: token, segment, and position. The token embedding contains a vector representation for each token representing the relation of this token with other tokens in the text. The segment embedding is used to distinguish the vector representation for two sentences in a sentence pair. The position embedding is used to specify the position of words in the text sentence. Thus, the embedding representation (Ei) is the concatenation of token, segment and position embeddings.

• The BERT Transformer Blocks takes the embedding representation (Ei) as input, and calculates a score that represents the contextualized value for each specific token in relation to all other tokens. This score is represented as (Ri) for each token in the processed sentence.

3. Classification & Post-Processing: In this step, the approach takes each predicted BERT representation (Ri) as input and feeds them into a dense layer with a softmax activation function. This layer obtains a BIO label for each token in the sentence, calculating a probability P for each label using the softmax function, as follows: (1) Pl|Ri=SoftmaxWoRi+bo

where the label l belongs to the set of BIO labels to be predicted. In addition, Wo is a matrix of weight parameters and bo is a bias vector, both learned by the dense layer. Finally, the special tokens ’[CLS], [SEP], [PAD]’ are removed to obtain the final BIO labels at the end of post-processing step.

Experimentation and Results

In this section, we describe experiments carried out to evaluate the proposed approach for negation and uncertainty detection. We will first describe the evaluation methodology, then the experiments that were carried out, followed by the results that were obtained.

Evaluation methodology

The evaluation methodology depends on the dataset used, as follows:

• NUBES corpus: this corpus was split by their authors into three subsets: training (75%), developing(10%), and testing (15%). Models trained with the NUBES corpus were executed just once. The testing subset was used to calculate the performance metrics.

• IULA corpus: this corpus does not provide an explicit division for training and testing subsets. Therefore, in this case we followed a cross-validation strategy with k = 5. The performance was calculated as the average of all five folds executed by the cross-validation strategy.

To evaluate the performance of the proposed approach, we used the following standard metrics: Precision (P), Recall (R), and F-score (F1). The F-score is calculated as a weighted average of the Precision and Recall measurements. A token is correctly classified when the predicted label is equal to the label indicated by the annotated corpus. The performance for the cue identification task and the scope recognition task is analyzed separately. (2) Precision=Number of tokens correctly predictedNumber of predicted tokens

(3) Recall=Number of tokens correctly predictedNumber of tokens in the dataset

(4) F-score=2∗Precision∗RecallPrecision+Recall

Experiments

The BiLSTM-CRF model proposed by Huang, Xu & Yu (2015) was used as baseline system to perform negation and uncertainty detection. Next, we performed the following experiments:

1. Experiment 1: Spanish biomedical embeddings proposed by Soares et al. (2019) were added to the BiLSTM-CRF model. The goal of this experiment was to analyze the impact of adding biomedical embeddings to the BiLSTM-CRF model.

2. Experiment 2: the goal of this experiment was to test the impact of using embeddings trained on clinical notes written in Spanish. Consequently, in-house clinical embeddings section ‘BiLSTM-CRF’ were added to the BiLSTM-CRF model.

3. Experiment 3: the goal of this experiment was to analyze the impact of using multilingual BERT embeddings to perform negation and uncertainty detection in Spanish. To perform this experiment, we used the BERT model as described in section ‘Bidirectional Encoder Representation for Transformers (BERT)’.

Validation

The Cancer dataset described in section ‘Datasets’ is used for validating the performance of trained models on the NUBES corpus. We used these models for validation because the NUBES corpus contains annotations for both negation and uncertainty (the IULA corpus cannot be used as it does not contain uncertainty annotations). This validation aims to measure the performance of trained models on the NUBES corpus to predict negation and uncertainty in a new dataset.

Implementation and Hyperparameters setting

To perform the previously explained experiments Python 3.7, TensorFlow (https://www.tensorflow.org/?hl=es-419) and Keras (https://keras.io/) were used. For the BiLSTM-CRF model the following parameters were settled: learning rate as 0.001, dropout as 0.5, the number of epochs was set to 60, the BiLSTM hidden size was set to 300, and the batch size to 512. For the BERT model, the fine-tuning was performed with a sequence length of 256 tokens, a batch size of 64, and five epochs. These values were established after training the models different times, and checking the best performance for these parameters. Data and code of the proposed approach can be found in GitHub (https://github.com/solarte7/NegationAndUncertainty).

Results

This section shows first the support and the performance metrics for each BIO label. Once the results of each label are shown, we will also describe the global performance for cue identification and scope recognition. We will conclude the section showing the validation results for the Cancer dataset.

Table 5 depicts the results obtained for each BIO label using the BiLSTM-CRF model and biomedical embeddings in the NUBES corpus (Lima Lopez et al., 2020). These kinds of tables have also been calculated for each model.

Table 5 Results for each BIO label using the NUBES corpus (BiLSTM-CRF + Biomedical Embeddings).

Label name	P	R	F1	Support	
-PAD-	1.0	1.0	1.0	474126	
B-NEG	0.95	0.93	0.94	1423	
B-NSCO	0.93	0.91	0.92	1322	
B-UNC	0.86	0.84	0.85	400	
B-USCO	0.84	0.79	0.81	400	
I-NEG	0.89	0.86	0.87	120	
I-NSCO	0.92	0.88	0.90	3901	
I-UNC	0.85	0.75	0.80	168	
I-USCO	0.82	0.77	0.79	1513	
O	0.98	0.99	0.98	41977	

The values of Table 5 will be used later to calculate the performance for cue detection and scope recognition as the weighted average between the “B” and “I” labels. In particular, the performance of the negation cue detection is obtained as the weighted average between “B-NEG” and “I-NEG” labels. On the other hand, the negation scope is obtained as the weighted average between “B-NSCO” and “I-NSCO”. In the same way, we calculate the performance for uncertainty detection labels. This procedure has been followed for each model and for each corpus.

Moreover, results are obtained using partial and exact match, as follows:

• Partial match: Refers to the performance at token level. When a cue or scope contains several tokens, the performance for each token is independently measured. For instance, in the sentence ”La biopsia nomuestra células cancerígenas”, the scope annotated in the corpus is underlined. However, the model could predict just the tokens “muestra” and ”células”. In this case, these tokens are considered as correctly predicted. In contrast, the token ”cancerígenas” is considered as an error as it does not match with the annotated token in the corpus.

• Exact match: Refers to the performance at cue and scope level. An exact match is then only considered in the case when the complete set of tokens predicted by the model corresponds to the set of tokens annotated by the experts in the corpus. In the above example, the scope is considered a correct prediction if the model predicts all the tokens annotated in the corpus (muestra células cancerígenas)”, otherwise, it is considered an error.

The following shows the results of experiments for cue identification, scope recognition, and validation with the Cancer dataset.

Cue identification

Table 6 shows the results obtained for the cue identification task in the experiments previously described. These results show the feasibility of both BiLSTM-CRF and BERT models to perform negation and uncertainty cue identification in clinical texts written in Spanish. As can be seen, the best performance was obtained in the third experiment in which the model was trained using multilingual BERT. Using the NUBES corpus, this model obtained an F-score of 95% and 84% for negation and uncertainty detection, respectively. While for the IULA corpus, the model obtained an F-score of 92% for negation detection.

Table 6 Results for cue identification (Partial match).

	Negation	Uncertainty	Negation	
	P	R	F1	P	R	F1	P	R	F1	
BiLSTM-CRF	0.86	0.82	0.83	0.79	0.76	0.77	0.82	0.78	0.80	
BiLSTM-CRF + Biomedical Embbedings	0.94	0.92	0.93	0.85	0.81	0.83	0.91	0.90	0.90	
BiLSTM-CRF + Clinical Embbedings	0.93	0.91	0.92	0.84	0.80	0.82	0.90	0.88	0.89	
Multilingual BERT	0.95	0.93	0.95	0.86	0.83	0.84	0.92	0.93	0.92	

In addition, Table 6 shows that when the BiLSTM-CRF model is combined with biomedical and clinical embeddings, it obtains competitive results in the cue identification task. In the first experiment, the BiLSTM-CRF model obtained an F-score of 93% and 83% for negation and uncertainty, respectively. In the second experiment an F-score of 92% and 82% were obtained. These results suggest that using biomedical and clinical embeddings is a useful approach to improve the performance of the BiLSTM-CRF model to detect negation and uncertainty in Spanish clinical texts. Moreover, using biomedical and clinical embeddings also improved the performance of the BiLSTM-CRF model in the IULA corpus.

According to Table 6, negation detection showed better performance than uncertainty detection. In the first experiment, the BiLSTM-CRF model obtained an F-score of 93% for negation detection and 83% for uncertainty detection. Meanwhile, models trained using multilingual BERT obtained an F-score of 95% for negation detection and 84% for uncertainty detection. Thus, these results highlight the fact that uncertainty detection is more difficult than negation detection in clinical texts written in Spanish.

The cue identification task measured using the exact match also shows competitive results. The best performance was also achieved using multilingual BERT. For the NUBES corpus, this model obtains an F-score of 95% and 83% for negation and uncertainty, respectively. The BiLSTM-CRF combined with biomedical embeddings obtains an F-score of 92% and 80% for negation and uncertainty. These results are similar to those shown in Table 6. This is because most of the negation and uncertainty cues contain only one annotated token, and consequently, the partial and exact matches have similar behavior.

Scope recognition

Table 7 describes the results obtained in the scope recognition task using a partial match. These results show the feasibility of deep learning-based methods to perform the scope recognition task for both negation and uncertainty detection. The best performance was obtained by using the BERT model. A 92% F-score for negation and 80% F-score for uncertainty were obtained using the NUBES corpus. Using the IULA corpus, this model obtained an F-score of 89% for negation detection.

Table 7 Results for scope recognition (Partial match).

	NUBES corpus	IULA corpus	
	Negation	Uncertainty	Negation	
	P	R	F1	P	R	F1	P	R	F1	
BiLSTM-CRF	0.84	0.76	0.79	0.72	0.69	0.70	0.77	0.74	0.75	
BiLSTM-CRF + Biomedical Embbedings	0.92	0.89	0.90	0.82	0.77	0.79	0.88	0.84	0.86	
BiLSTM-CRF + Clinical Embbedings	0.92	0.87	0.89	0.81	0.75	0.78	0.87	0.83	0.85	
Multilingual BERT	0.93	0.90	0.92	0.82	0.79	0.80	0.91	0.86	0.88	

The BiLSTM-CRF model combined with biomedical and clinical embeddings also showed a competitive performance in the scope recognition task. In the first experiment, it obtained an F-score of 90% for negation and 79% for uncertainty detection using the NUBES corpus. In the second experiment, 89% and 78% were obtained for negation and uncertainty, respectively. Results obtained suggest that adding clinical and biomedical embeddings increases the ability of the BiLSTM-CRF model to perform the scope recognition task. Results obtained in the IULA corpus also show that using biomedical and clinical embeddings has a positive impact on the performance of the BiLSTM-CRF model.

Table 7 shows that negation detection performs better than uncertainty detection in experiments carried out with the NUBES corpus. This suggests that extracting the uncertainty scope is more difficult than extracting the negation scope. This behavior can be explained by the fact that negation also performs better than uncertainty in the cue identification task. Therefore, the difficulties in extracting uncertainty cues can also affect scope recognition, since cue identification and scope recognition are related tasks.

Table 8 describes the results obtained in the scope recognition task measured with an exact match. Using the NUBES corpus, the best performance was obtained with the BERT model. This model obtained an F-score of 88% for negation and 72% for uncertainty detection, respectively. These results show a decrease in the performance compared to those results from partial matches. The decrease percentage is 4% for negation and 8% for uncertainty detection, respectively. Regarding the IULA corpus, the BERT model obtained an F-score of 86%, which is 2% lower compared to results from partial matches. The performance decrease can also be seen in the BiLSTM-based models. The performance decrease is due to the fact that the scope commonly consists of several tokens in the sentence. Therefore, correctly predicting all annotated tokens by experts is a difficult task. Note that the decrease percentage is smaller in negation detection than uncertainty detection. This behavior can be explained by the following facts:

Table 8 Results for scope recognition (Exact match).

	NUBES corpus	IULA corpus	
	Negation	Uncertainty	Negation	
	P	R	F1	P	R	F1	P	R	F1	
1. BiLSTM-CRF	0.81	0.73	0.76	0.67	0.63	0.64	0.74	0.72	0.73	
2. BiLSTM-CRF + Biomedical Embbedings	0.88	0.85	0.86	0.73	0.70	0.71	0.84	0.83	0.84	
3. BiLSTM-CRF + Clinical Embbedings	0.86	0.84	0.84	0.71	0.68	0.69	0.84	0.80	0.82	
4. Multilingual BERT	0.90	0.86	0.88	0.75	0.70	0.72	0.89	0.84	0.86	

• Predicting the exact scope when it finishes at the end of the sentence is easier than predicting the scope when it ends in any other part of a sentence. Moreover, negation has more scopes that finish with the end of the sentence than uncertainty. Predicting these scopes generates fewer errors than predicting scopes that do not finish at the end of the sentence. In the NUBES corpus, 27% of the negation scopes finish at the end of the sentence. On the other hand, only 7% of the uncertainty scopes finish at the end of the sentence. In the IULA corpus (annotated only with negation), 33% of the scopes finish with the sentence.

• Predicting the scope in a short sentence is easier than predicting it in a long sentence. The percentage of sentences whose number of tokens is below the first quartile (Q1) is higher in negation than in uncertainty. We consider these sentences as short sentences. For instance, the sentence “Paciente sindolor toráxico.” (patient without chest pain.) is considered as a short sentence, and the scope is shown underlined. In the NUBES corpus, 19% of sentences annotated with negation are below Q1, and only 7% of the sentences annotated with uncertainty are below Q1.

• Predicting the scope in a long sentence produces more errors than predicting it in a short sentence. The percentage of sentences whose number of tokens is above the third quartile (Q3) is greater in uncertainty than in negation. In the NUBES corpus, 52% of sentences annotated with uncertainty are above Q3. On the other hand, only 30% of the sentences annotated with negation are above Q3.

Transferring results to the cancer dataset

Table 9 shows obtained results for the validation process with the Cancer dataset using a partial match. These results show the performance of trained models with the NUBES corpus for predicting cues and scopes on the Cancer dataset. Table 9 shows that the best performance was obtained by using the BERT model. In the cue identification task, this model obtained an F-score of 90% and 82% for negation and uncertainty, respectively. In the scope recognition task, the BERT model obtained an F-score of 87% and 78% for negation and uncertainty, respectively. In addition, the performance of the BiLSTM-CRF model combined with biomedical and clinical embeddings also showed competitive results. For instance, the model that uses clinical embeddings obtained an F-score of 89% and 80% in cue identification for negation and uncertainty, respectively. In the scope recognition task, this model obtained an F-score of 86% and 77% for negation and uncertainty, respectively.

Table 9 Validation results with the Cancer dataset (Partial match).

	Cue detection	Scope recognition	
	Negation	Uncertainty	Negation	Uncertainty	
	P	R	F1	P	R	F1	P	R	F1	P	R	F1	
BiLSTM-CRF + Biomedical Embbedings	0.89	0.87	0.88	0.75	0.80	0.78	0.86	0.83	0.84	0.79	0.74	0.76	
BiLSTM-CRF + Clinical Embbedings	0.91	0.88	0.89	0.80	0.81	0.80	0.87	0.85	0.86	0.79	0.76	0.77	
Multilingual BERT	0.91	0.89	0.90	0.84	0.80	0.82	0.89	0.86	0.87	0.79	0.78	0.78	

Although results from Table 9 show a lower performance compared to those described in Tables 6 and 7, results are still promising. The validation process showed the ability of the deep learning-based models trained with the NUBES corpus to predict negation and uncertainty in a different dataset. This fact suggests that models trained with the NUBES corpus can be used to detect negation and uncertainty in new clinical texts.

On the other hand, Table 10 shows results for validation with the Cancer dataset using an exact match. These results also show that in the cue detection task the performance is similar to those results from partial matching. However, in the scope detection task, a decrease in the performance can be seen compared to those results obtained in a partial match. The BERT model obtained an F-score of 84% and 74% for negation and uncertainty, respectively. The decrease percentage was 3% for negation and 4% for uncertainty detection.

Table 10 Validation results with the Cancer dataset (Exact match).

	Cue detection	Scope recognition	
	Negation	Uncertainty	Negation	Uncertainty	
	P	R	F1	P	R	F1	P	R	F1	P	R	F1	
BiLSTM-CRF + Biomedical Embbedings	0.87	0.86	0.87	0.75	0.80	0.78	0.81	0.80	0.80	0.71	0.69	0.70	
BiLSTM-CRF + Clinical Embbedings	0.91	0.88	0.89	0.80	0.81	0.80	0.82	0.83	0.75	0.72	0.71	0.71	
Multilingual BERT	0.91	0.89	0.90	0.82	0.80	0.81	0.85	0.84	0.84	0.75	0.73	0.74	

If we compare the performance of the proposed approach with other studies in the literature, we found that for the cue identification task, this approach obtains competitive results compared to those reported by Lima Lopez et al. (2020), as shown in Table 11. A direct comparison with other proposals such as the one in Santiso et al. (2018) is not possible because it reports results for negated entities within the scope, but not for negation cues. Other proposals such as Santiso et al. (2020), Costumero et al. (2014) and Cotik et al. (2016) use private corpora which are not publicly available so a direct comparison is also not possible. Moreover, those proposals focused only on negation detection, but not on uncertainty.

Table 11 Comparison with other proposals in the cue identification task.

	Cue detection - NUBES corpus	
	Negation	Uncertainty	
	P	R	F1	P	R	F1	
Lima Lopez et al. (2020)	0.96	0.95	0.95	0.87	0.83	0.85	
Our approach (BiLSTM-based)	0.94	0.92	0.93	0.85	0.81	0.83	
Our approach (BERT-based)	0.95	0.95	0.95	0.86	0.83	0.84	

In the scope recognition task, the proposed approach outperforms previous studies both for negation and uncertainty detection in Spanish. Table 12 shows the performance of the proposed approach in comparison with other proposals for the scope detection task. If one further analyzed the results from the NUBES corpus, the approach presented in this paper outperforms the Lima Lopez et al. (2020) proposal in both tasks (improvement of 2% in the case of uncertainty and 2% in the case of negation) using a partial match. If one analyzes the IULA corpus, the approach presented in this paper outperforms the results reported in Santiso et al. (2018) and Zavala & Martinez (2020) by 3% and 1%, respectively. In the IULA corpus, the performance was compared using an exact match. In addition, the approach proposed in this paper presents the following advantages over other studies:

Table 12 Comparison with other proposals in the scope recognition task (F-score).

	NUBES corpus	IULA corpus	
Proposal	Negation	Uncertainty	Negation	
Santiso et al. (2018)	–	–	0.83	
Zavala & Martinez (2020)	–	–	0.85	
Lima Lopez et al. (2020)	0.90	0.78	–	
Our approach (BiLSTM-based)	0.90	0.79	0.84	
Our approach (BERT-based)	0.92	0.80	0.86	

• The proposed approach takes advantage of transfer learning techniques, biomedical word embeddings, and pre-trained models to automatically represent text features using dense vector representations. These representations are used for negation and uncertainty detection. In contrast, other proposals such as Lima Lopez et al. (2020); Santiso et al. (2018) require a considerable set of hand-crafted features to obtain comparable results. The process of manual feature extraction can be time-consuming and costly. Moreover, the proposal described in Zavala & Martinez (2020) depends on the syntactic properties of a sentence and other word-level annotated features to perform negation detection. On the contrary, the approach proposed in this paper only uses suitable biomedical word embeddings and contextual embeddings without depending on any other annotated features.

• The proposed approach deals with negation and uncertainty detection in a single step, improving those approaches (Zavala & Martinez, 2020; Santiso et al., 2018; Santiso et al., 2020; Costumero et al., 2014; Cotik et al., 2016) that only deal with negation detection in Spanish clinical texts.

• This approach improves performance in the scope recognition task in comparison with approaches Lima Lopez et al. (2020) dealing both with negation and uncertainty in Spanish, as one can see in Table 12.

Discussion

The proposed approach has shown the feasibility of deep learning-based methods to perform negation and uncertainty detection from clinical texts written in Spanish. We found that both BiLSTM-CRF and BERT models obtained competitive results for both tasks: cue identification and scope recognition. The use of biomedical embeddings for Spanish and contextualized embeddings from multilingual BERT results in an improvement of the negation and uncertainty detection process. The proposed approach automatically represents text features using word embeddings and contextual embeddings, and uses them to detect uncertainty and negation. This is an advantage over previous proposals that require complex hand-crafted rules (Costumero et al., 2014; Cotik et al., 2016; Solarte-Pabón, Menasalvas & Rodriguez-González, 2020) or a considerable set of hand-crafted features (Lima Lopez et al., 2020) to obtain comparable results.

Obtained results showed that scope recognition is a more complex task than cue identification. This could be because in most cases negation and uncertainty cues consist of a single token. However, the scope frequently contains a longer sequence of tokens which makes it more difficult to detect it properly. As Table 7 shows, the proposed approach in this paper obtained competitive results for the scope recognition task, outperforming previous studies available for this task (Santiso et al., 2018; Lima Lopez et al., 2020).

The proposed approach performs better negation detection than uncertainty detection. Specifically, in the cue identification task, the best performance showed an F-score of 95% and 84% for negation and uncertainty detection, respectively (See Table 6). This behavior can be explained by two facts:

• The number of annotations with negation is higher than with uncertainty in the NUBES corpus. In this corpus, there are more than 7,500 sentences annotated with negation and only 2,219 sentences annotated with uncertainty, which affects the training of the models.

• Negation cues have less variability than uncertainty cues. Only five negation cues are used in the NUBES corpus (“no, sin, negativo, negativos, niega”) to express 87% of all negation annotations. However, the set of words used to express uncertainty is much broader. The five more frequent uncertainty cues (“probable, posible, compatible con, sospecha de, parece”) are used only in less than 48% of all uncertainty annotations. This fact once again affects the training process, reducing the performance of the models to perform uncertainty detection.

Although both BiLSTM-CRF and BERT models have shown feasibility to perform negation and uncertainty detection in Spanish, BERT obtains better results than BiLSTM-CRF in some specific cases. In particular, we observed that the BERT model tends to learn the scope better in those cases in which it appears before the cue, such as in the sentence:

“Mujer con cáncer de pulmón, HER2negativo” (Woman with lung cancer, HER2 negative).

In this sentence, the scope is underlined and the negation cue is in bold. In this case, the BERT model is able to learn that the scope is before the cue. The BiLSTM-CRF model failed in most of these cases. The BERT model also showed better performance than the BiLSTM-CRF model in those cases where the scope appears in both directions of the cue, as in the following sentence:

“TAC cerebralnegativopara células tumorales.” (Negative brain CT scan for tumor cells.)

In this sentence, we can see that the scope (underlined) is before and after the cue (showed in bold). In these cases, the BERT model recognizes the scope in both directions (after and before the cue). On the other hand, the BiLSTM-CRF model tends to recognize only the part of the scope that appears after the cue.

The BERT model also outperforms the BiLSTM-CRF model when predicting labels with fewer annotations in the annotated corpus. In particular, the NUBES corpus contains more than 6,500 annotations for syntactic negation, and only 460 for lexical negation. In the syntactic negation, both BiLSTM-CRF and BERT models perform accurately. However, for lexical negation, the BiLSTM model fails to recognize some lexical cues with a low number of annotations in the corpus, while the BERT model recognizes them properly.

In ‘Related works’, we analyzed that syntactic properties of the sentence have been used in rule-based approaches for negation detection. Nevertheless, these properties can also be used in the context of deep learning methods and particularly for sequence-labeling tasks. In the context of negation and uncertainty detection in Spanish, several proposals (Lima Lopez et al., 2020; Zavala & Martinez, 2020) have used syntactic properties such as Part of Speech (POS) tagging and dependence relations between words as input features to the deep learning models. However, in the approach presented in this paper, we propose using suitable biomedical and clinical embeddings for the Spanish language instead of syntactical properties, and we have shown that this achieves competitive results. However, it will be interesting to explore additional syntactic properties in future works to improve the performance of negation and uncertainty detection in Spanish.

The deep learning-based approach proposed in this paper relies on the availability of annotated corpora which is a time-consuming task. Fortunately, for the Spanish language, in recent years, great efforts have been made to create manually annotated corpora (Lima Lopez et al., 2020; Marimon, Vivaldi & Bel, 2017). We want to acknowledge the efforts to annotate these public corpora that have made it possible to propose and validate new approaches such as the one we have presented in this paper. For those languages in which annotated corpora are not available, researchers have to dedicate a large amount of time to obtain a manually annotated corpus.

Despite the promising results in this study, there are still some limitations that need to be addressed. In particular, we identify the following causes of error which can affect the performance of the proposed approach in both: BiLSTM-CRF and BERT-based models:

• Those cues that appear rarely are not detected. These cases mainly occur in uncertainty detection where some syntactic cues such as “Versus” and “O” (or), are annotated as uncertainty cues. However, the NUBES corpus has very few annotations using these words which means the approach does not recognize them as cues. As a consequence, the approach fails in sentences such as these examples: (Cue shown in bold):

– “Cáncer pulmonar Versus inflamación del lóbulo derecho”. (Lung cancer Versus right lobe inflammation).

– “Carcinoma estadio IV o estadio IIIB”. (Stage IV or stage IIIB carcinoma.)

• In the scope recognition task, most errors are caused by discontinuous scopes. This occurs when the sequence of tokens affected by a negation cue or an uncertainty cue are separated in different positions of the text sentence. In the next example, the scope (shown underlined) is discontinuous. We show the sentence annotated in the gold standard compared to that predicted by the system.

Gold: ”Paciente parcialmente orientado (si en tiempo, noen espacio)” (Partially oriented patient (in time, not in space)

Predicted: ”Paciente parcialmente orientado (si en tiempo, noen espacio)”

Another case of discontinuous scopes occurs when a sentence contains a sequence of cues, but some tokens in the sentence are not affected by any cue. A sequence of cues is more frequent in negation detection, as one can see in the following examples:

i) Gold: “Nohipertensión arterial , novómito, nosangrado.” (No high blood pressure, no vomiting, no bleeding.)

i) Predicted:“Nohipertensión arterial, novómito, nosangrado.”

ii) Gold: “No dolor, fiebre alta desde la noche anterior, no sangrado, no vomito.” (No pain, high fever since last night, no bleeding, no vomiting.)

ii) Predicted: “No dolor, fiebre alta desde la noche anterior, no sangrado, no vomito.”

Note that in the first sentence (i), each token is affected by a negation cue. In this case, the approach correctly predicts the scope for each cue. However, in the second example (ii), the tokens “fiebre alta desde la noche anterior” are not negated, but the approach classifies them as negated. A possible cause of error when predicting discontinuous scopes may be associated with the fact that these scopes are not frequent in the annotated corpora. For instance, in the NUBES corpus, only about 5% of annotated scopes are discontinuous. However, these cases should be analyzed further in the future to improve performance.

Conclusions

In this paper, we proposed a deep learning-based approach for negation and uncertainty detection in clinical texts written in Spanish. Results obtained have shown the ability of deep learning methods to perform both cue identification and scope recognition tasks. The proposed approach takes advantage of transfer learning techniques, word embeddings, and pre-trained models to automatically represent text features, thus avoiding the time-consuming feature engineering process. This approach is useful for medical text mining applications because it can recognize negation and uncertainty in a single step. Moreover, the approach improves previous studies, as has been shown.

Both BiLSTM and BERT models have shown promising results for negation and uncertainty detection in clinical texts written in Spanish. The BERT model performed better than the BiLSTM model in some specific cases. In the cue identification task, the BERT model performed better at predicting labels that have fewer annotations in a corpus. In the scope recognition task, this model worked better at extracting the scope that appears before the cue, or when it appears on both sides of the cue. These facts increased the performance of the BERT model over the BiLSTM-CRF model.

In this paper, we also conducted a validation process using real-life clinical data which showed promising results. Deep learning models trained on the NUBES corpus have shown the ability to predict negation and uncertainty in datasets that are different from those on which they were trained. In future studies, we will explore more transformer-based architectures to perform negation and uncertainty detection.

Additional Information and Declarations

Competing Interests

Author Contributions

Data Availability

The authors declare there are no competing interests.

Oswaldo Solarte Pabón conceived and designed the experiments, performed the experiments, analyzed the data, performed the computation work, prepared figures and/or tables, authored or reviewed drafts of the paper, and approved the final draft.

Orlando Montenegro conceived and designed the experiments, performed the experiments, performed the computation work, prepared figures and/or tables, and approved the final draft.

Maria Torrente, Alejandro Rodríguez González and Mariano Provencio analyzed the data, authored or reviewed drafts of the paper, and approved the final draft.

Ernestina Menasalvas conceived and designed the experiments, performed the experiments, prepared figures and/or tables, authored or reviewed drafts of the paper, and approved the final draft.

The following information was supplied regarding data availability:

The data and code of the proposed approach are available at GitHub: https://github.com/solarte7/NegationAndUncertainty.

Two datasets were used for training our models: NUBES and IULA. They are available at:

- http://eines.iula.upf.edu/brat/#/NegationOnCR_IULA

- https://github.com/Vicomtech/NUBes-negation-uncertainty-biomedical-corpus.

The Cancer dataset contains real-life data from patients treated with lung and breast cancer in “Hospital Puerta de Hierro” in Madrid, (Spain). This dataset is governed by the General Data Protection Regulation (GDPR) (EU) 2016/679 of the European Parliament. The Cancer dataset is available upon request and can be accessible for research purposes after an evaluation by the hospital’s ethics committee. Researchers can request a copy of the Cancer dataset by contacting Dr. Cristina Avendaño Solá, Director of the hospital’s ethics committee at Hospital Universitario Puerta de Hierro (Spain): cristina.avendano@uam.es.

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
