# Peer review of "Negation and uncertainty detection in clinical texts written in Spanish: a deep learning-based approach"

_PeerJ Computer Science, doi:10.7717/peerj-cs.913_

## Round 0.1 · original submission · Major Revisions

One of the reviewers has some major suggestions for the paper although, in general, comments are quite positive. Please prepare a new version according to these suggestions, as well as a report explaining how are you going to address these comments.

·

Basic reporting

The authors describe an approach for the detection of negation and uncertainty detection in clinical texts written in Spanish using deep learning. The authors have used two deep learning approaches, namely BiLSTM-CRF and BERT. The paper builds on various similar works the exploit deep learning approaches and annotated data for Spanish and advance the state-of-art in an incremental way. Although, the paper does not present any entirely new break though idea, in my view, it is still of general interest. The authors demonstrate that either better or competitive results can be achieved with the approaches based on deep learning models on both negation and uncertainty detection in clinical texts.

Experimental design

The experimantal design is, on the whole, well done and the paper requires relatively minor modifications. Some issues that could be addressed are mentioned below.

Non-exploitation of syntactic properties:
In Section 2.1 (lines 149-150) the authors mention some works that exploit syntactic properties of the sentence to extract the scope (e.g. Cotik et al., 2016; Zhou et al., 2015; Peng et al., 2018). Although this was used in the context of rule-based approaches, a question arises regarding whether this would not be useful also for approaches that use deep learning. I recommend that the authors address this issue in e.g., 5 Discussion.

Deep learning approaches require effort of labelling data:
I tend to agree with the authors that creating rules manually can be a lengthy process (Section 2.1, lines 151-152). I also agree that feature engineering used in many machine learning approaches can be a time-consuming task too. For instance, you mention that NUBES corpus includes 29,682 labelled records. I imagine that labelling this data must also be a time-consuming task. Although, in the case presented in this paper this data is already available, this might not not be the case if this solution were applied to other languages, for which such labelled corpus is not available. So, I find it unfair that the effort on labelling large amounts of training data has not been mentioned by the authors. Future evaluations and comparisons of different systems should take into account not just F1, but various other costs, such as those associated with preparation of auxiliary labelled data, features or rules manually. But this is future work. As for this paper, can you please mention that the solution presented is not without extra costs?

Explain better how the labels are dealt with the deep learning system:
Section 3.2.1 discusses the model BiLSTM-CRF and this is accompanied by Figure 2. Similarly, Section 3.2.2 describes the BERT model and this is accompanied by Figure 3. In both figures we can see how one particular sentence (i.e. “La biopsia no muestra células cancerígenas” in case of BiLSTM-CRF) is fed in. At the beginning of Section 3.2. (lines 328-330) the authors show how one sentence is annotated : ([’Paciente : O’, ’Sin : B-NegCue’, ’dolor : B-NegScope’, ’tora´xico : I-NegScope’, ’. : O ’].).
From the description it is not clear how the labels are handled. Are they fed also into BiLSTM-CRF and BERT, or just “attached”? This should be clarified.

Improve the description of the BERT representation:
The authors mention that they use three embeddings, token, segment and position embeddings. As for the token embedding, the authors use the symbol E_n to represent it. From the context it seems that the other two embeddings are represented using symbols E_1 and E_2. This could be clarified.
The authors then discuss the output representation (R_1, R_2, R_n). The equation (1) should make it clear that the index i spans the values 1,2 and n - if I interpreted this correctly. So, as we can see, each word representation in this scheme is associated with a particular label, so this probably answers my doubt raised in the previous paragraph (i.e., how the labels are handled).
But what are the exact values of w_0 and b_0 used in the implementation?

Evaluation in the scope task
Section 4.1 discusses the evaluation measures used, which are pretty standard. It appears that the result of a particular scope task can have a binary result determining whether the system did this right or not. However, scope task involves determining all elements in the scope. So, for instance, when considering your example sentence in Section 3.2 (lines 328-330), i.e., ([’Paciente : O’, ’Sin : B-NegCue’, ’dolor : B-NegScope’, ’tora´xico : I-NegScope’, ’. : O ’].), we see that there are two elements within the NegScope. It could happen that the system identifies just one of them correctly. Would, in this case, this be considered as an error? This point could be explained better. Also, other measures could be used in case the answer is only partially correct.

When is the WordPiece Tokenization activated?
This issue is discussed in Section 3.2.2. and an example is given of how some words are separated into morphological units. For instance, “biopsia” is represented as “bio ##psia”. But what mechanism determines which token should be (or not) decomposed this way? Whey, for instance, is the word “muestra” (or células) nor decomposed in a similar manner?

Validity of the findings

The paper and and the results presented are of general interest. It shows how some existing techniques can be adapted to new settings and improve the results of the state-of-the art approaches.

Additional comments

Minor points on presentation and wording

Table 2 summarizing the datasets includes two different kinds of information (on size and on cues and scope). It seems that it would be better to separate this into two tables.

Section 2.2, line 157:
in the second step those features are trained into a classifier to perform predictions =>
the second step those features are used to train a classifier to perform predictions

Section 2.3, line 193:
from syntactic trees with a convolutional layer that’s features are concatenated =>
from syntactic trees with a convolutional layer whose features are concatenated

Figure 1:
The labels NegCue and NegScope appear in rectangles that are shaded. If this is printed on a black-and-while printer, the labels are hardly legible, as the shading is too dark.

Title of Section 4.5.3 Validation Results
The title “Transfer results” would seem clearer, given the contents

The reference Santiso et al., (2018) in incomplete

Reviewer 2 ·

Basic reporting

In the Introduction section, the motivation of the work focuses on the improvement of deep learning approaches in several nlp tasks, but not on negation and speculation detection. There are some works that show improvements in negation and speculation tasks, and this should also be included as motivation for your work. The following are some of the works focused on this topic:
-Zavala, R. R., & Martinez, P. (2020). The Impact of Pretrained Language Models on Negation and Speculation Detection in Cross-Lingual Medical Text: Comparative Study. JMIR Medical Informatics, 8(12), e18953.
-Britto, B. K., & Khandelwal, A. (2020). Resolving the Scope of Speculation and Negation using Transformer-Based Architectures. arXiv preprint arXiv:2001.02885.
-Fei, H., Ren, Y., & Ji, D. (2020). Negation and speculation scope detection using recursive neural conditional random fields. Neurocomputing, 374, 22-29.
-Al-Khawaldeh, F. T. (2019). Speculation and Negation Detection for Arabic Biomedical Texts. World of Computer Science & Information Technology Journal, 9(3).
-Dalloux, C., Claveau, V., & Grabar, N. (2019, September). Speculation and negation detection in French biomedical corpora. In RANLP 2019-Recent Advances in Natural Language Processing (pp. 1-10).
In the Related works section you present some of them, but you should indicate the differences and similarities between your work and the existing ones.

The negation overview in which all existing corpora in all languages are compiled is missing.
Jiménez-Zafra, S. M., Morante, R., Teresa Martín-Valdivia, M., & Ureña-López, L. A. (2020). Corpora annotated with negation: An overview. Computational Linguistics, 46(1), 1-52.

Typos:
- The sentence “In these cases, the scope recognition fails because all tokens in a sentence can be taken as the scope.” appears twice. Lines 146-148.
- BiLSMT ---> BiLSTM (Line 578).

Experimental design

In the description of the datasets it would be convenient to add the total number of negation and speculation cues of each corpus.

Regarding the Cancer dataset, as it is the first time it is presented, it should be included the number of annotators and the inter-annotator agreement for both, negation and speculation. How did you resolve the disagreement cases?

Why did you explore BiLSTM-CRF and BERT for negation and uncertainty detection? This should be justified in subsection 3.2. This is very important as it will make the research rigorous and not a simple test of algorithms.

There is one part that is not clear to me. If I understand correctly, the detection of cues and scopes is modeled jointly. How do you then know which scope corresponds to which cue?

In order to the approach be replicable, it must be indicated which probability values have been used to determine the predicted labels (B-Cue, I-Cue, O, B-CueScope,...).

Validity of the findings

Comparison with other proposals regarding cue detection should also be included, as it has been done for scope identification in Table 6.

The discussion section is very interesting because it helps to learn how the algorithms studied work and the advantages and shortcomings of each one. When talking about limitations that need to be addressed, do these limitations correspond to BiLSTM, BERT or both? This should be clarified in the paper. On the other hand, when it is said that most of the scope errors are due to discontinuous scopes, some examples of annotation of discontinuous scope are presented but how the system has labeled those scopes should also be included, so that we can see where it is failing the system.

Additional comments

The paper deals with a very interesting and important topic that affects all natural language processing tasks. Moreover, it does so in Spanish, a language less studied in this subject. In general, a good job has been done, but the aspects that I have indicated in the previous sections should be addressed to improve the paper before publication. Therefore, my recommendation is major revisions.

---

## Round 0.2 · accepted · Accept

All the reviewers consider the paper is ready for publication, and so do I.

·

Basic reporting

The paper discusses the problem of detecting negation and uncertainty in clinical texts in Spanish.
The topic is general and should be of interest to others.
The section in Related work presents a comprehensive account of current work.
The methodology is described in a clear manner and the results obtained are competitive and of interest.
The authors have responded well to my previous comments.
My recommendation is to accept the paper.

Experimental design

No additional comment

Validity of the findings

No additional comment

Additional comments

Some suggestion to improve the formulation:
Line 252:
model, which results showed => model whose results showed
Line 286:
The results obtained will show => The results obtained show
Lines 411-412:
On the other hand, the BERT model is based on transformer architectures which also have been improved sequence-labeling tasks in the biomedical domain => ??
On the other hand, the BERT model is based on transformer architectures which also has been improved for sequence-labeling tasks in the biomedical domain

Figure 2: The arrow below Marix T’ and Matrix L’ in the direction of Matrix H should be moved. It should join Matrix L’ and Matrix H.

Reviewer 2 ·

Basic reporting

The authors have taken into account the reviewers' suggestions and have made the appropriate changes, improving the article. They have done a great job. Therefore, I recommend its acceptance but there are two details to be addressed.

Typos:

Line 84: allow to advances —> allow to advance
Line 243: (Zavala and Martínez, 2020) —> Zavala and Martínez (2020)

Please, add to Table 4 the Total of negation and uncertainty cues.

Experimental design

-

Validity of the findings

-